# Clinical Features and Outcomes of the Association of Co-Infections in Children with Laboratory-Confirmed Influenza during the 2022–2023 Season: A Romanian Perspective

**DOI:** 10.3390/v15102035

**Published:** 2023-09-30

**Authors:** Mădălina-Maria Merișescu, Monica Luminița Luminos, Carmen Pavelescu, Gheorghiţă Jugulete

**Affiliations:** 1“Matei Balş” National Institute for Infectious Diseases, No. 1, Calistrat Grozovici Street, 2nd District, 021105 Bucharest, Romania; madalina.merisescu@umfcd.ro (M.-M.M.); luminita.luminos@umfcd.ro (M.L.L.); 2Faculty of Medicine, University of Medicine and Pharmacy “Carol Davila”, No. 37, Dionisie Lupu Street, 2nd District, 020021 Bucharest, Romania; carmen.pavelescu@rez.umfcd.ro

**Keywords:** COVID-19, SARS-CoV-2, children, influenza

## Abstract

The 2022–2023 influenza season in Romania was characterized by high pediatric hospitalization rates, predominated due to influenza A subtypes (H1N1) pdm09 and H3N2. The lowered population immunity to influenza after the SARS-CoV-2 pandemic and the subsequent stoppage of influenza circulation, particularly in children who had limited pre-pandemic exposures, influenced hospitalization among immunosuppressed children and patients with concurrent medical conditions who are at an increased risk for developing severe forms of influenza. This study focused on the characteristics of influenza issues among pediatric patients, as well as the relationship between different influenza virus types/subtypes and viral and bacterial co-infections, as well as illness severity in the 2022–2023 season after the SARS-CoV-2 pandemic. We conducted a retrospective clinical analysis on 301 cases of influenza in pediatric inpatients (age ≤ 18 years) who were hospitalized at the National Institute of Infectious Diseases “Prof. Dr. Matei Balș” IX Pediatric Infectious Diseases Clinical Section between October 2022 and February 2023. The study group’s median age was 4.7 years, and the 1–4 year age group had the highest representation (57.8%). Moderate clinical forms were found in 61.7% of cases, whereas severe versions represented 18.2% of cases. Most of the complications were respiratory (acute interstitial pneumonia, 76.1%), hematological (72.1%, represented by intra-infectious and deficiency anemia, leukopenia, and thrombocytopenia), and 33.6% were digestive, such as diarrheal disease, liver cytolysis syndrome, and the acute dehydration syndrome associated with an electrolyte imbalance (71.4%). Severe complications were associated with a risk of unfavorable evolution: acute respiratory failure and neurological complications (convulsions, encephalitis). No deaths were reported. We noticed that the flu season of 2022–2023 was characterized by the association of co-infections (viral, bacterial, fungal, and parasitic), which evolved more severely, with prolonged hospitalization and more complications (*p* < 0.05), and the time of use of oxygen therapy was statistically significant (*p* < 0.05); the number of influenza vaccinations in this group was zero. In conclusion, co-infections with respiratory viruses increase the disease severity of the pediatric population to influenza, especially among young children who are more vulnerable to developing a serious illness. We recommend that all people above the age of six months should receive vaccinations against influenza to prevent the illness and its severe complications.

## 1. Introduction

After the start of the COVID-19 pandemic in December 2019, there were notable decreases in influenza activity throughout the world [1,2,3,4]. Numerous research has demonstrated that the epidemiological and clinical characteristics of respiratory viral illnesses have changed, especially in children, and the most common causes of morbidity and mortality in children under the age of five are viral lower respiratory tract infections, especially pneumonia and bronchiolitis brought on by influenza and RSV [5,6,7]. Influenza A and B viruses cause seasonal influenza in humans of all ages, with a variety of severity levels [7]. Children may be even more at risk of developing severe influenza during rebound seasons due to having fewer prior exposures and the changes in the age patterns of first exposure, since they have fewer total lifetime influenza exposures and are often exposed during the first few years of life. Reduced influenza vaccination rates and a gap in influenza circulation have reduced population immunity, which could increase the severity of influenza epidemics [6,7,8]. Children are more likely than adults to develop symptoms of influenza and carry a disproportionate amount of the disease’s burden, with annual incidence rates reaching up to 30% [8]. Due to the high incidence of influenza-related hospitalizations among children aged 6 to 59 months, the World Health Organization identifies this age category as one of the risk groups for influenza and underlines the importance of prioritizing them for yearly influenza vaccines [9]. Furthermore, because of their lack of past exposure and immunity to the virus, children are typically the principal transmitters of influenza in the community; they shed the virus at greater titers and for a longer length of time than adults do [9,10,11,12,13,14,15].

When comparing the severity of influenza in children of different ages, the clinical symptoms of the illness, such as fever, malaise, cough, rhinitis, headache, etc., as well as the consequences, vary due to the virus type and subtype [16,17,18,19]. These variables make it extremely difficult to distinguish between them clinically and to start effective management based on contemporary medical care to lower morbidity [20,21,22,23]. To help clarify the issue, we conducted a study that compared the clinical forms (easy, moderate, and severe) based on the age groups, clinical symptoms, and treatment in a pediatric population with influenza and influenza with co-infections. The relevance of these respiratory disorders and the clinical manifestation of viral co-infections are more likely to be found in children who have been diagnosed with influenza A + B viruses, who have more severe clinical symptoms, and need hospitalization more frequently and for a long time [22,23,24]. Children who have chronic diseases and other medical conditions are more likely to experience complications from influenza than healthy children [24]. Several pulmonary and extrapulmonary problems, such as myocarditis and pericarditis, myositis or rhabdomyolysis, and seizures, might occur [24,25,26,27,28].

In the EU countries, including Romania, influenza virus activity nearly reached pre-pandemic levels during the 2022–2023 influenza season. Compared to the four preceding seasons, this one was distinguished by an earlier start to the seasonal epidemic and an earlier peak in positivity based on national surveillance data. According to the Romanian National Center for Statistics, the influenza vaccine administration in the pediatric population decreased to 2.5% during the 2022–2023 season [29]. Everyone over the age of six months in Romania is eligible for the influenza vaccination, but the Ministry of Health only offers it complementarily to vulnerable populations like patients with coexisting conditions or HIV infection, pregnant women, people older than 65, institutionalized individuals, and social and medical professionals [29,30].

In the present study, we aim to analyze the epidemiological particularities of influenza in children in the post-pandemic season of COVID-19 (the type/subtype of the circulating influenza virus, the age groups and gender of the infected children, and co-infections associated with influenza) in unvaccinated children. We will also identify the clinical forms of the disease, and the complications of influenza in children who were hospitalized between October 2022 and February 2023. The information gathered may be essential in predicting disease progression, prognosis, and treatment plans, particularly in pediatric populations.

## 2. Methods

To achieve the proposed objectives, we conducted a clinical retrospective study of influenza laboratory-confirmed pediatric cases, hospitalized between October 2022 and February 2023 in the Clinical Section IX Pediatric Infectious Diseases of the National Institute of Infectious Diseases “Prof. Dr. Matei Balș”, Bucharest, Romania. The analyzed data included the following parameters: age, sex, the type/subtype of the influenza virus, the clinical features of the disease, the complications, and the duration of hospitalizations. To date, influenza detections were higher for influenza A H1N1pdm09 than H3N2, and the largest proportion of hospitalized cases among children was between the ages of 0 and 4 years old. We also analyzed the co-infections associated with influenza for the children included in the study, as well as their impact on the disease. We defined co-infections as the association of one or more bacterial, viral, and fungal laboratory-positive tests in addition to influenza. Patients with at least one influenza diagnosis and a positive finding for the respiratory specimen (nasopharyngeal or pharyngeal swab) from laboratory tests were identified as having laboratory-confirmed influenza via a reverse transcription polymerase chain reaction (RT-PCR). Real-time RT-PCR was used to detect influenza A and influenza B (without lineage) viruses using the GenXpert instrument from Cepheid in Sunnyvale, California, and the AllplexTM Respiratory Panel 1 kit from Seegene in Seoul, Republic of Korea. An internal assay was used to perform genetic subtyping on samples that tested positive for influenza B.

### 2.1. Case Identification

All consecutive pediatric patients aged 0–18 years, who were referred to our Pediatric Department, were categorized as having mild, moderate, or severe influenza based on their body temperature (<38.5, 38.5–39.0, and >39.0) in °C and their duration of hospitalization (<7 days, between 7 and 14 days, and >14 days). Severe forms of influenza were assessed for children treated in the intensive care unit, with invasive mechanical ventilation and oxygen administration to maintain a saturation of at least 92%, or for whom extrapulmonary complications occurred. The moderate form criteria were established for children who had been confirmed to have upper respiratory tract illness and had pneumonia confirmed via an X-ray, and for whom hypoxemia was present. Children with influenza symptoms who tested positive for influenza but did not meet the criteria for moderate to severe influenza were classified as having an easy form of influenza.

The clinical course was defined as follows (see Table 1).

A positive influenza viral PCR result also showed the presence of one or more bacterial pathogens, such as was referred to as a bacterial co-infection; the most common co-infecting species was *Streptococcus pneumoniae* (33%), while a wide range of other pathogens caused the remaining infections. Within the first 48 h of admission, bacterial cultures were acquired from the blood, genuine sputum, and bronchoalveolar fluid. A pathogenic microorganism that was isolated within 14 days of the beginning of influenza was considered a concurrent infection. *Streptococcus pyogenes*, *Haemophilus influenzae*, *Klebsiella pneumoniae*, and *Mycoplasma pneumoniae* were among the additional bacteria that were found to be responsible for co-infections.

### 2.2. Detection of Routine Blood Parameters

A routine blood analyzer was used for the detection of the usual laboratory investigations (hemogram, biochemical tests, and inflammatory tests); the lymphocyte count, monocyte count, and platelets were recorded. In all cases, studies were performed, as well as, depending on the complications, cultures (blood cultures, pharyngeal exudate, urine culture) and paraclinical investigations (cardio-pulmonary radiography). Any unfavorable clinical episode within 14 days of a diagnosis of influenza was considered to be an influenza-related complication and included respiratory, hematological, and digestive complications, but also neurological.

The multiplex polymerase chain reaction test FilmArray—the BioFire Blood Culture Identification Panel (BCID)—is an additional novel diagnostic technique. In roughly one hour, it can find 27 targets, including Gram-positive and Gram-negative bacteria as well as yeast.

The study and the informed consent forms from all parents and legal guardians of the patients included in this study were approved by the Bioethics Committee of the National Institute for Infectious Diseases “Prof. Dr. Matei Balș”, Bucharest, Romania, according to the Helsinki Declaration of 1964.

### 2.3. Statistical Analysis

Data were determined for normal distribution using the Kolmogorov–Smirnov test. Continuous variables were presented as medians with interquartile ranges (IQRs) and were compared using the Mann–Whitney test.

The differences in means and medians for the continuous variables were provided using Student’s t-test or the Kruskal–Wallis test. Categorical variables were analyzed univariately using Fisher’s exact test. The median age of the cases and controls was compared using the Wilcoxon rank-sum test. Statistics were considered significant for *p*-values under 0.05. All statistical analyses were performed using GraphPad 9.1.1.

## 3. Results

Between October 2022 and February 2023, a total number of 301 consecutive pediatric cases were hospitalized at the National Institute for Infectious Diseases “Prof. Dr. Matei Bals”, Bucharest, Romania, with laboratory-confirmed influenza A (244/301, 81.06%; H1N1 pdm09 represents 66.4% and H3N2 = 33.6%), influenza B (24/301, 7.97%), influenza A + B, (19/301, 6.31%), and not-subtyped influenza (14/301, 4.65%). In the influenza season of 2022–2023, we diagnosed a spread with a peak number of 108 pediatric cases in December 2022 (35.88%) and 78 in January 2023 (25.91%). In this study, influenza A viruses were the most dominant strain type during most months and caused 244 episodes, which represented 81% of the total group. The proportion of children with influenza A, subtype H1N1 pdm09, was 66.4%, and 33.6% were recorded for A/H3N2. (See Table 2). The proportion of girls was higher than that of boys (58.14% vs. 41.86). All cases evolved favorably, with no deaths throughout the study.

The median age was 4.7 years (interquartile range (IQR) of 2.8–12.2 years). Children infected with A/H1N1 pdm09 were detected in 18.85% of cases, and 9.84% had A/H3N2 in the 7–12 age category. Infection with influenza viruses A + B showed the highest percentage (31.58%), represented by the age category of 0–1 year. Regarding influenza B, the best-represented age category was between 13 and 18 years old, with a percentage of 29.17% (see Figure 1, Figure 2 and Figure 3).

There were significant differences in the age distribution based on the influenza type. The median age of children with influenza A and B infections was 3.338 and 6.228 years, respectively (*p* < 0.001). The median age of children with a dual influenza A and B infection was lower than the other groups at 3.21 years old (*p* < 0.05).

The demographics and clinical characteristics of the pediatric patients in the study group are shown in Table 1. Younger children were more likely to be symptomatically infected with influenza A than influenza B, with an average age of cases of 3.338 years for influenza A and 6.228 years for influenza B (*p*< 0.05). There were significant differences in the age distribution based on the subtype or co-infection of influenza A + B and for age groups. (See Table 3, Figure 1, Figure 2 and Figure 3).

At presentation, patients with influenza A had a higher admission temperature (median of 38.5 °C, IQR [37.7–38.8]) vs. 37.6 (36.8–38.2]; *p* = 0.02) for influenza B, and 100% of patients with influenza A + B had a high fever (>38.8 °C). Other baseline vital signs and clinical data were similar between the pediatric patient groups. In a comparison of the demographic characteristics and clinical outcomes, we found that hospital stays for severe forms were more frequently required for those aged under 1 year old for influenza A (2.4% vs. 0.62%, *p* < 0.001) than in patients with influenza B infection. All of the admitted infants who were younger than one year old were full-term births, they all had siblings, and their medical history was not relevant.

There was a significant difference between the influenza subtypes in terms of the clinical symptoms and days of antiviral administrations. The respiratory symptoms, except for rhinorrhea and a cough, were statistically significant. There were differences in the clinical features between A(H3N2) and A(H1N1) pdm09 and between the B and A + B influenza co-infection for the respiratory and digestive symptoms. Fever, cough, and dyspnea were the three most common symptoms, with statistical differences between the two types of influenza, A and B, between subtypes of influenza A, and when we compared the influenza A group with A + B co-infection (see Table 1). More patients with influenza A reported respiratory symptoms, while influenza A + B and influenza B reported more frequent digestive manifestations, such as vomiting and diarrhea. For details, see Table 3.

The median number of days of hospitalization was found to be statistically significant for comparison between influenza A and A + B, and between the A and B types. The longest median number of days of hospitalizations was found for co-infection A + B influenza (10 days). The mean length of hospital stay was 5.4 ± 5.5 days [4–14 days], with a median of 8 days for influenza B. Oseltamivir therapy was begun for 80.07% of the entire group. At admission, 84.02% of patients with influenza A and 83.33% of cases with influenza B received oseltamivir if they had moderate or severe forms. For A + B influenza, we administrated oseltamivir for 84.21% of pediatric patients with severe forms. Vomiting during oseltamivir administration was reported in 10 patients, and 24 patients reported diarrhea, but the treatment was not discontinued because the patients evolved favorably.

Patients with influenza A + B had a significantly longer length of hospital stay (10 days, [7,8,9,10,11], *p* = 0.011), although they had a longer administration of oseltamivir than the other groups (7 days, IQR [7,8,9], *p* = 0.05).

A significantly greater proportion of patients with moderate influenza were identified, and some of them were reported to have extra-pulmonary complications. Neurological and digestive manifestations of the moderate influenza form were found in hospitalized pediatric patients and were significantly different between the moderate and severe groups. Although, significant differences in respiratory involvement were identified. Analyses adjusting for patients diagnosed with pneumonia, bronchiolitis, and laryngitis demonstrated comparable results. (See Table 4).

In 79 (26.2%) of the hospitalized children’s influenza cases, we observed the association of other infections: bacterial (11.96%), viral (5.9%), and fungal (8.3%) (see Figure 4).

Fifty-nine patients (19.6%) of the total group had co-infections. Bacterial and fungal co-infections were most common in severe forms of influenza. Bacterial co-infections were reported for 23 patients, with *S. aureus* as the most common (39.5%), followed by *M. pneumoniae* (31%), *H. influenza* (13.25%), and others. Fungal co-pathogens were reported for *Aspergillus* spp. (38%), *Candida* spp. (29%), *Candida albicans* (*n* = 23.3%), and others (9.7%). Respiratory syncytial virus (RSV) type A was found in 12 cases: four patients in the age group of 0–1 year with the severe form, and eight cases of the moderate form in the 2–3-year-old age group (*p* < 0.05).

None of the patients received influenza vaccines within the previous year before admission to our pediatric department.

## 4. Discussion

Due to the abandoning of preventive techniques, we observe an increase in the frequency of influenza in children after a time with a relatively low number of influenza cases in our country. The clinical characteristics, treatment, co-infections, and days of hospitalization in pediatric patients with influenza A, B, and A + B during the 2022–2023 influenza season were examined in this study. According to the influenza virus type and subtype, the clinical characteristics and outcomes demonstrate relevant changes. Among children between 0 and 1 year old, influenza A + B virus infection appeared to be more serious than influenza A or B virus infection. Influenza A + B has low incidence but is associated with an increased risk of influenza complications. The prevalence of influenza complications was influenced by bacterial, fungal, and viral co-infection, necessitating the use of advanced medication. Patients with influenza B had atypical manifestations, which included more frequent digestive symptoms, and the age category was statistically significant for the 13–18 age group. Influenza A was described according to influenza A strains, and patients with H1N1 pdm09 were shown to be younger than those with H3N2. This result is consistent with findings from past research that found that young children and infants have an increased risk of being admitted to hospital with influenza. We identified several possible independent risk factors: age group, represented by the 0–1 year-olds and 13–18-year-olds; and respiratory symptoms, such as fever, cough and dyspnea, vomiting, and myalgia. In a study of pediatric patients with a confirmed influenza infection, 95% of patients reported fever, 77% reported coughing, and 26% reported headache and myalgia.

Abdominal pain, nausea, and diarrhea, as gastrointestinal symptoms [31], were mentioned less frequently, at only 7%.

Co-infections with RSV type A were most frequently involved in these cases, especially at younger ages. Therefore, in co-infections with RSV and influenza viruses, it is challenging to determine whether each virus directly contributes to the pathogenesis of complications in babies that are of 0–2 years of age. Respiratory and neurological complications associated with influenza have been reported in numerous studies. We identified that the children with neurological involvement were healthy children before the influenza event, and they did not experience consequences after the episode, even though half of them were in the 2–3-year-old age group [32,33].

In terms of incidence and severity, influenza continues to be a risk and can cause complications such as pneumonia, which is one of the most common side effects of influenza, or even severe diseases such as encephalitis in the pediatric population. The purpose of the influenza vaccine is to lessen the morbidity, hospitalization days, and death caused by this illness, as well as its complications. And in this regard, it is crucial to consider the requirements for immunizing the population of healthy youngsters [34,35].

As a limitation of our study, we were unable to identify a comparison group of patients with severe complex influenza who had received the vaccination, since all of the children in our trial had not received the seasonal influenza vaccine [36].

Immunization may protect against both mild and severe complex influenza; these important findings emphasize the value of vaccination for children aged >6 months, especially for children who have siblings, as we found that the presence of older siblings was a risk factor for influenza-confirmed admission in the age group under 2 years old.

## 5. Conclusions

To reduce influenza-related hospital admissions among young children, the immunization of children who are eligible for the influenza vaccine and who have siblings should be strongly encouraged during the influenza season, because if influenza returns, it is unknown how the virus will continue to evolve.

## Figures and Tables

**Figure 1 viruses-15-02035-f001:**
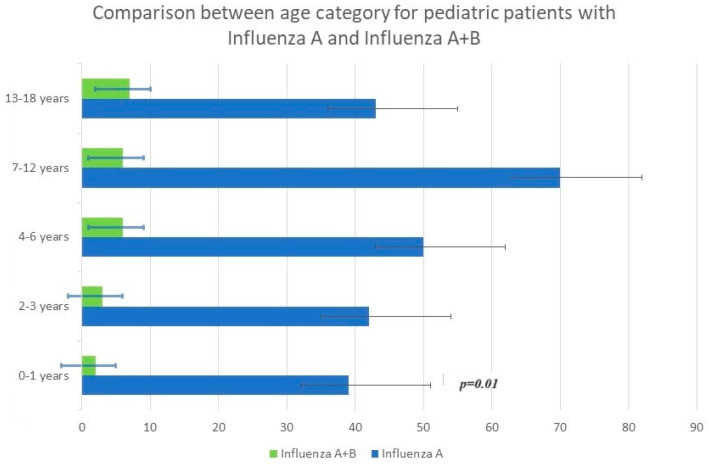
Influenza incidence rate in the pediatric population, season of 2022–2023, in Romania. Age distribution of A and A + B influenza cases (Wilcoxon rank-sum test).

**Figure 2 viruses-15-02035-f002:**
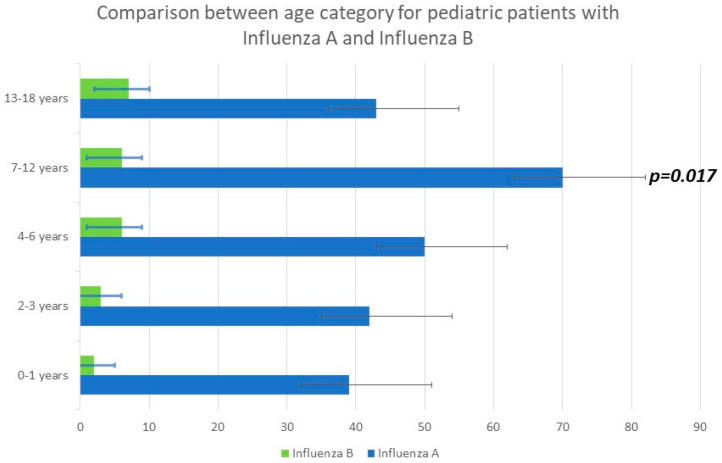
Comparison of influenza incidence rate, by age, between influenza A and B. (*p* = 0.017).

**Figure 3 viruses-15-02035-f003:**
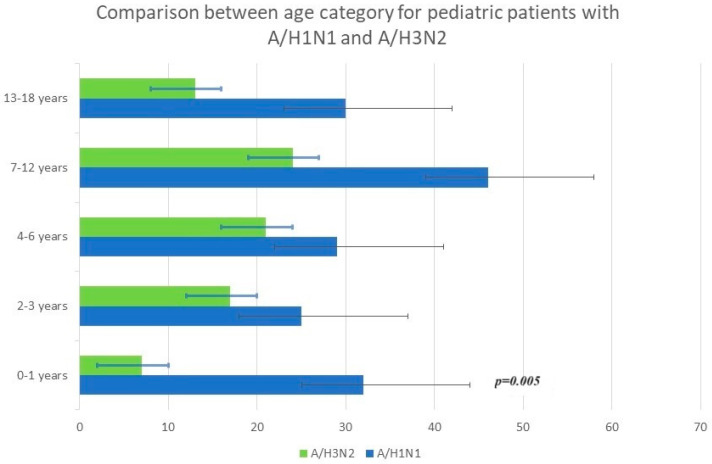
Comparison by age category for influenza A/H1N1 and A/H3N2.

**Figure 4 viruses-15-02035-f004:**
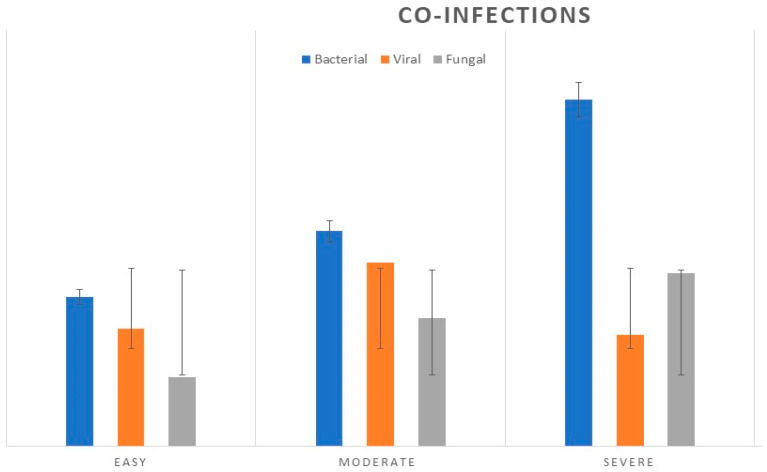
The number of cases of laboratory-confirmed bacterial, viral, and fungal co-infections in pediatric cases diagnosed with influenza in the 2022–2023 season, compared based on clinical forms.

**Table 1 viruses-15-02035-t001:** Clinical course.

Category	Easy Form	Moderate Form	Severe Form
Body temperature	<38.5 (+)	38.5–39 (++)	>39 (+++)
Duration of hospitalization	<7 days (+)	7–14 (++)	>14 days (+++)
ICU	(−)	(−)	ICU admission (+)
Upper respiratory tract infection	(+)	(++)	(+++)
Pneumonia (Chest X-ray)	(−)	(+)	(+)
Oxygen saturation	SpO_2_ > 95% (+)	SpO_2_ 92–95% (++)	SpO_2_ < 92% (+++)
Lower respiratory tract illness	(−)	(+)	(++)
Extrapulmonary complications	(−)	(+)	(++)
Score (+)	0–4	5–11	12–18

**Table 2 viruses-15-02035-t002:** The number of patients in the study groups based on influenza type.

Category	Total	Oct-22	Nov-22	Dec-23	Jan-23	Feb-23
Not subtyped	14	0	1	9	3	1
Influenza A and B	19	0	0	10	8	1
Influenza Type B	24	0	3	8	4	9
Influenza Type A	244	2	5	108	78	51

**Table 3 viruses-15-02035-t003:** Comparison of clinical outcomes based on influenza virus type/subtype in the pediatric population; *p* * = comparison between influenza A (H3N2) and A (H1N1) pdm09. *p* ** = comparison between influenza A and A + B. *p* *** comparison between influenza A and B. The *p* values displayed are for the comparison between the influenza type/subtypes, and were compared using the Mann–Whitney test.

Parameter	Influenza A Group (*n* = 244)				Influenza A + B (*n* = 19)		Influenza B (*n* = 24)	
	Total	A(H3N2) (*n* = 33.6%)	A(H1N1)pdm09 (*n* = 66.4%)	*p* *	*p* **	Total	Total	*p* ***
Gender *n* (%)								
Male	135 (55.33)	46 (18.85)	89 (36.48)	1.000	0.112	11 (57.9)	11 (45.83)	0.289
Female	109 (44.67)	36 (14.75)	73 (29.91)	0.05	0.03	8 (42.1)	13 (54.16)	0.233
Age median [IQR], years		3.478 [2.6–6.2]	3.338 [2.28–7.125]	0.1	0.005	3.21 [1.91–4.33]	6.228 [5.13–9.29]	0.001
Age group in years at influenza infection *n* (%)								
0–1 years		7 (2.87)	32 (13.11)	0.005	0.01	6 (31.58)	2 (8.33)	0.15
2–3 years		17 (6.97)	25 (10.25)	0.7	0.12	2 (10.53)	3 (12.5)	0.112
4–6 years		21 (8.6)	29 (11.89)	0.075	0.07	3 (15.79)	6 (25)	0.5
7–12 years		24 (9.84)	46 (18.85)	0.25	0.5	3 (15.79)	6 (25)	0.25
13–18 years		13 (5.33)	30 (12.3)	0.125	0.5	5 (26.32)	7 (29.17)	0.017
Clinical symptoms *n* (%)								
Fever	240 (98.36)	81 (33.2)	159 (65.16)	0.0001	0.001	19 (100)	22 (91.67)	0.626
Respiratory symptoms	226 (92.7)	89 (36.48)	137 (56.15)	0.029	0.05	11 (94.5)	18 (75)	0.566
Cough	165 (83.2)	58 (82.2)	107 (84.8)	0.15	0.03	10 (84.6)	14 (58.33)	0.027
Rhinorrhea	210 (65.7)	47 (66.1)	163 (61.7)	0.06	0.093	13 (68.42)	10 (41.67)	0.071
Sore throat	189 (77.46)	56 (22.95)	133 (54.5)	0.189	0.9	12 (63.16)	8 (33.33)	0.303
Dyspnea	68 (27.87)	23 (9.43)	45 (18.44)	0.001	0.567	5 (26.32)	6 (25)	0.005
Gastrointestinal symptoms *n* (%)								
Vomiting	149 (61.07)	108 (44.26)	41 (16.8)	0.246	0.04	7 (36.84)	21 (87.5)	0.05
Abdominal pain	151 (61.89)	63 (25.82)	88 (36.07)	0.327	-	0 (0)	4 (16.67)	1.000
Diarrhea	99 (40.57)	30 (12.3)	69 (28.29)	0.599	0.001	3 (15.79)	8 (33.33)	0.016
Other symptoms Myalgia	143 (58.6)	46 (18.85)	97 (39.75)	0.233	0.112	7 (36.84)	8 (33.33)	0.017
Treatment *n* (%)								
Antiviral therapy Oseltamivir	205 (84.02)	42 (76.2)	163 (66.8)	<0.001	0.023	16 (84.21)	20 (83.33)	0.05
Days of hospitalization, *n* = numbers od days and [IQR]		5 [5–6]	7 [7–8]	0.2	0.01	10 [7–11]	8 [7–9]	0.012
Days of antiviral administrations (Oseltamivir), *n* = (number of days), [IQR].		5 [4–5]	6 [5–7]	0.1	0.05	7 [7–9]	5 [5–6]	0.2

**Table 4 viruses-15-02035-t004:** Complications between easy, moderate, and severe influenza forms in the pediatric patients. (* Comparison between the moderate and severe influenza groups).

Variable	Total (*n* = 301) 100%	Easy (*n* = 60) 20%	Moderate (*n* = 185) 61.7%	Severe (*n* = 56) 18.2%	*p*-Value *
Gender *n*, (%) male/female	126 (41.86)/175 (58.14)	28 (46.6)/32 (53.3)	75 (40.5)/110 (59.46)	23 (41.07)/33 (58.92)	0.048
ENT involvement	41 (13.62)	18 (5.5)	17 (13.6)	6 (23.9)	0.5
Sinusitis	21 (6.98)	7 (2.33)	9 (2.99)	5 (1.66)	0.1
Acute otitis media	20 (6.64)	11 (3.65)	8 (2.66)	1 (0.33)	0.33
Respiratory involvement	93 (30.9)	14 (23.3)	36 (19.4)	43 (76.1)	0.05
Interstitial Pneumonia	27 (8.97)	4 (2)	6 (1.99)	17	0.01
Secondary Bacterial Pneumonia	26 (8.64)	3 (0.996)	10 (3.32)	13 (4.32)	0.011
RSV co-infection	12 (4)	0	8 (2.66)	4 (2)	0.05
Fungal co-infections	11 (3.65)	1 (0.33)	3 (1)	7 (2.33)	0.01
Bronchiolitis obliterans	22 (7.31)	1 (0.33)	11 (3.654)	10 (3.32)	0.05
Laryngitis	18 (5.98)	6 (1.99)	9 (2.99)	3 (0.996)	0.03
Neurological manifestations	6 (2)	1 (0.33)	4 (2.0)	1 (0.33)	0.01
Febrile seizure	2 (0.664)	0	2 (0.664)	0	0.012
Acute encephalomyelitis	3 (0.996)	2 (0.664)	1 (0.33)	0	0.1
Encephalitis	1 (0.33)	0	0	1 (0.33)	0.13

## Data Availability

The datasets generated and analyzed during the current study are available from the corresponding author upon request.

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
