# Peer review of "Clinical Features and Outcomes of the Association of Co-Infections in Children with Laboratory-Confirmed Influenza during the 2022–2023 Season: A Romanian Perspective"

_viruses, 2023, doi:10.3390/v15102035_

Round 1

Reviewer 1 Report

The paper presented by Meriá¹£escu et al. provides a clear and well-structured description of the clinical features and outcomes of influenza infections in children for 5 consecutive months of the 2022-2023 season. The authors found cases of simultaneous influenza A+B infections, as well as various cases of coinfections with bacteria, fungi, and parasites. The paper highlights that coinfections had a statistically significant association with age and the influenza subtype.

Overall, the paper is well-written and easy to read. However, there are a few issues that need to be addressed.

1) Abstract
- The influenza H1N1 subtype should be corrected to A(H1N1)pdm09, as this is the correct spelling of the current H1N1 virus. This correction should be made throughout the text.
- In the abstract (line 32), the authors state that the frequency of coinfections in 2022-2023 was higher than in previous seasons. However, they do not provide any data for the previous season in the main text. This discrepancy should be corrected.

2) Introduction
- The sentence in lines 60-62 appears to be a fragment and needs to be revised for clarity.

3) Methods
- The methods for laboratory confirmation of coinfections should be added to provide a comprehensive understanding of the study's methodology.
- Additionally, it would be helpful to include the criteria used to categorize patients as having easy, moderate, or severe influenza.

4) Results
- In line 129, the phrase "the most dominant type during most seasons" is incorrect since the paper only presents data from one season. It would be more appropriate to use "months" instead of "seasons" in this context.
- Figure 1 is exhaustive and could be simplified. I suggest removing it and including only the table that is currently placed under the figure. Additionally, please add a "Total" column on the right side of the table.
- Figure 2 should be redone. The age groups are discrete, so a stretched graph is not appropriate. I recommend converting it to a histogram or using another type of graph that allows for clear visualization of separate age groups.

Addressing these issues will further enhance the clarity and quality of the paper.

Author Response

we appreciate everything you suggested us to modify.
We have attached the document for the answers.
Thank you for your consideration and suggestion,

We hope to accept the manuscript, with the changes.

Reviewer 2 Report

Comments for the author of Viruses manuscript viruses-2608022:

The author of the Viruses manuscript “Clinical features and outcomes of association of co-infections in children with laboratory-confirmed influenza, during 2022-2023 season: a Romanian perspective”, present their work in the clinical evaluation of influenza virus infections in the pediatric population in Romania.  Specifically, they are interested in this population that had lower immunity against influenza virus infection due to minimal circulation of this virus in the first year of the SARS-CoV-2 pandemic.  Their study focused on patients that were less than 18 years of age and in an inpatient setting.  They analyzed 301 cases in this study that lasted from October 2022 through February 2023.  Their main findings are that there was a range of complications, including respiratory, hematological, digestive, and neurological.  They did not observe any deaths in this study.  This group observed more co-infections during the season evaluated than they had seen in other seasons, and these included viral, bacterial, fungal, and parasitic.  None of the patients in this study had been vaccinated.  Based on these findings, the authors propose that co-infections make the disease worse and that children older than six-months-old should get vaccinated.  Below are some comments that I would like the authors to address as they revise the manuscript.   

General Comments:

  1. In the abstract (Line 35-36), it was unclear how co-infections can increase the severity of the pediatric population’s immunity to influenza.  It can increase the disease severity but maybe not immunity severity.
  2.  Many of the figures (2-6) did not have y-axis labels.   This was especially problematic with Figure 5 where not only was it unclear what was being presented. 
  3. Similarly, there was a lot of text taken up with numbers that appeared to be presented in the figures as well.  It might be easier if the text complemented the date presented in the figures rather than repeated it.

Please review sentence structure.

Author Response

Dear reviewer and colleague, please accept the changes we have made, we hope to that we have managed to bring the required changes to the standards required for publication. please accept our response to your requests and thank you for them.

Thank you, 

Reviewer 3 Report

General points: 

  • Avoid using the term ‘flu’ instead of ‘influenza’. Also, ‘Influenza’ refers to the disease and ‘influenza’ refers to the virus so correct accordingly. 

  • Why has the study included data from Oct 2022-Feb 2023 and not the full northern hemisphere season (October till May)?

  • For the methods section, it would be clearer if this was divided into several sections:

    • Case diagnosis and identification

      • Identification of laboratory-confirmed influenza by RT-PCR testing of respiratory specimens

      • Identification of type/subtype of influenza- how was this performed?  

      • Identification of concurrent infection – how were the pathogenic micro-organisms isolated?

      • Routine testing – hemogram, biochemical tests, inflammatory tests. What do these tests identify?

      • Identification of complications – cultures (blood cultures, pharyngeal exclude, urine culture), paraclinical investigations (cardio-pulmonary radiography) – what do these tests identify. 

  • Clinical manifestations and severity index

    • It would be helpful to have a table here detailing the different clinical manifestations and how these are linked to severity (can a severity index or scoring be applied?). This would make clear the distinction  between what is concluded in the manuscript (line 24) as ‘average clinical forms’ and (line 24) ‘severe versions’. 

  • Analysis and statistics

    • Explain which groups participants were divided into for analysis

    • Mention how the influenza incidence rate is calculated, the different statistical tests used and software used

  • Results: This can be divided into subheadings/subsections to make it easier to follow. The subheading should state the conclusion of the figures/tables described in that subsection. When describing the data in each section, start in a sequential manner describing each figure number in order to make the data presented easier to follow. Figures must have labelling on both X- and Y-axis. If statistical significance is indicated on figures, then groups which are statistically different must be annotated. All figure and table legends must state which statistical test was used to analyse the data.

Specific points:

Line 13: correct nomenclature ‘A/(H1N1)pdm09’ and ‘A(H3N2)’

Line 19: Should be ‘influenza types/subtypes’ 

Line 23: What is mean by ‘most significant age group’? Please specify

Lines 24-25: The manuscript does not define what is meant by ‘average clinical form’ and ‘severe forms’. There needs to be a breakdown table to define what these terms mean so that the conclusions are clear.

Line 31: ‘Frequent’ to be made ‘frequently’

Line 32: Is 16% a cumulative average number for ‘previous years’? Can the authors state which influenza seasons are being referred to here? Moreover, the comparison with previous years is not shown in the data in the manuscript so how can the authors conclude this in the abstract?

Line 34: Is this a typo ‘p > 0.05’, because the authors state that it is statistically significant

Line 34: The authors state that influenza vaccination in this group was zero, presumably meaning for the 2022-2023 season. What about co-infections groups from the previous years?

Line 36: The conclusion is that ‘coinfections with respiratory viruses increase the severity of the pediatric population’s immunity to influenza’. This sentence needs to be reworded. Their data shows that coinfections with respiratory viruses result in an increase in severe clinical outcomes; although it is likely that this is due to reduced immunity, there is no evidence to show that it is due to reduced immunity, so they shouldn’t mention the immune response in their conclusion. 

Line 37: At the start of the sentence use the words ‘We recommend’ because the authors don’t show any evidence in the manuscript that vaccination prevents illness and its severe complications.

Line 42: Instead of ‘notable falls’ use words ‘notable decrease’.

Line 44: Can this sentence be made more specific? Since which period of time have the epidemiological and clinical characteristics of respiratory illnesses changed?

Line 45: Where does reference 7 show evidence that the epidemiological and clinical characteristics of respiratory illnesses have changed?

Line 46: References are needed for the statement ‘Influenza A and B viruses cause seasonal influenza in humans of all ages with a variety of severity levels.

Line 49: References are needed for the statement ‘Reduced influenza vaccination rates and a gap in influenza circulation have reduced population immunity, which could increase the severity of influenza epidemics.’

Lines 53-56: WHO reference needed for this statement. 

Line 61: ‘clinical presentation’ to be changed to ‘clinical presentations’

Line 62: ‘vary’ to be changed to ‘varies’

Lines 62-64: This statement is unclear. What groups are you trying to differentiate between? Be more specific.

Lines 64-67: Reference needed for the statement here. Also it is not clear what is meant by the ‘relevance of these respiratory disorders are more likely found in children’. 

Lines 73-75: Reference needed for this statement. Also, elaborate this point. Was the peak significantly different compared to the previous four seasons?

Line 83: ‘type’ to be replaced with ‘type/subtype’

Line 84: Reword ‘gender,  of affected children’ to ‘gender of infected children’

Line 86: the statement ‘evolution of influenza’ is misleading because it can be interpreted as genetic evolution of the influenza virus.

Lines 87-88: ‘Illness development’ replace with ‘disease progression’

Line 94: ‘type’ to be replaced with ‘type/subtype’

Lines 94-95: ‘clinical form of the disease’ to be replaced with ‘clinical features’

Line 95: ‘evolution of the flu’ is an ambiguous term in this context, re-phrase it. 

Line 96: correct nomenclature ‘A/(H1N1)pdm09’ and ‘A(H3N2)’

Line 96: ambiguous statement: ‘influenza detections were higher for Influenza A H1N1 and H3N2’ – compared to what? And which one was higher A H1N1 or A H3N2?

Lines 95-96: The statement made here should be in the results section. 

Line 98: Elaborate how you analysed the impact of co-infections on the disease. 

Line 124-125: Correct nomenclature for H1N1 and H3N2 as before. Were the influenza B lineage typed? If yes, please state the lineage, if not, please comment that they were not.

Can the influenza A+B cases be further differentiated into influenza A(H1N1)pdm09 + B versus A(H3N2) + B?

Figures 2-4: Can figures 2-4 have all the different groups shown on a single graph? This would make it much clearer for the reader to do side-by-side comparisons and the Y-axis would be the same. Instead of shaded plots (which indicates area under curve analysis), different groups can be represented on a single graph with box and whisker plots showing the median and 25th and 75th percentiles. This would be more intuitive for the reader. The Y-axis needs to be labelled. Statistical significance between different groups can be compared with connecting lines between groups on the graph and reporting the p-value or stars which can then be explained in the legend. It is mentioned that Wilcoxon Rank sum test is used but this has not been mentioned in the methods.

Lines 136-144: The description would be made clearer if it was described what the graphs show X-axis versus Y-axis. Secondly, as mentioned above a single graph would be easier to describe. Why does this paragraph start with a description of Figure 4? Line 141: influenza ‘subtype’ to be replaced with ‘type’ as that is what is described in this sentence.

Table 1. The heading of table 1 needs to describe which statistical test has been used to analyse the data. Perhaps the data in the table can be heatmapped according to percentages, this would make the table easier to follow. The same goes for the P-values shown, these could be two different colours (one for significant values and the other for not significant values). 

Lines 166-168: the effects of oseltamivir treatment on recovery between different influenza types/subtypes have not been described here but instead are described in detail in Lines 180-189, so suggest not to mention oseltamivir in lines 166-168.

Line 180-181: State that you describe the median number of days for hospitalisations for oseltamivir treated patients.

Lines 180-189: The outcomes of antiviral therapy are not clear, because they are no untreated controls to compare against. Moreover, the authors do not describe which categories of patients have received antiviral treatment, i.e. ‘easy’, ‘moderate’ or ‘severe’ influenza cases?

Figure 5. Why is there a need to show this data when it is already shown in Table 1? Moreover, the Y-axis is not labelled.

Line 196-202: Define what is meant by ‘easy’, ‘moderate’ or ‘severe’ influenza. This can be done in the methods section. 

Table 2. Needs to explain which statistical test was used here. Heatmapping according to percentages and p-values significance would make the data easier to follow. 

Figure 6. This would be easier to follow if there was a category for ‘no co-infection’ as well. There is no labelling on the Y-axis. Also, Figure 6 can have extra graphs, Figure 6b-d, which represent a breakdown of the different species involved for b) bacterial, c) viral and d) fungal infections. 

Lines 216-217: Need to state which statistical test was used. 

Lines 246-247: Needs a reference for this statement. Used the word ‘previously’ before the statement ‘we identified…’

- Some inconsistent terminology used such as description of 'easy', 'moderate' and 'severe' forms of influenza is made confusing by inconsistent use of  the word 'complications' and 'clinical forms.' For e.g. are 'severe clinical forms' and 'severe complications' the same thing? Lines 24-29

- The authors sometimes say the opposite of what they mean for e.g. Lines 34-36. 

- Nomenclature for 'influenza' (virus) vs 'Influenza (disease) and the subtypes needs to be corrected 

Author Response

we hope that we have managed to bring the required changes to the standards required for publication. please accept our response to your requests and thank you for them.
